# The Mediating Role of Innovative Behavior on the Effect of Digital Leadership on Intrapreneurship Intention and Job Performance

**DOI:** 10.3390/bs13100874

**Published:** 2023-10-23

**Authors:** Murat Sagbas, Onur Oktaysoy, Ethem Topcuoglu, Erdogan Kaygin, Fahri Alp Erdogan

**Affiliations:** 1Ataturk Strategic Research Institute, National Defense University, Besiktas, Istanbul 34334, Turkey; muratsagbass@gmail.com (M.S.); falperdogan98@gmail.com (F.A.E.); 2Faculty of Economics and Administrative Sciences, Kafkas University, Merkez, Kars 36000, Turkey; onurkavak@kafkas.edu.tr (O.O.); kaygin@kafkas.edu.tr (E.K.); 3Academy of Civil Aviation, Giresun University, Merkez, Giresun 28200, Turkey

**Keywords:** innovative behavior, digital leadership, intrapreneurship intention, job performance

## Abstract

Depending on technological developments, digital transformation represents an inevitable reality for organizations. Based on this reality, digital leadership, which is a new understanding of leadership, has emerged. In accordance with the literature, digital leaders are expected to transform organizations under the leadership of innovation, thus encouraging high performance and efficiency. The present study aimed to measure the mediating effect of innovative behavior on the effect of digital leadership on job performance and intrapreneurship intention using data collected from 390 people working in the IT sector in Istanbul and a structural equation modeling method. The data obtained in this structural equation modeling study were analyzed in the Smart-PLS program. It is anticipated that the present study, in which the relationship between the variables is supported by various theories, will contribute to the extant literature. The results of this study indicate that innovative behavior has a fully mediating impact on the effect of digital leadership on intrapreneurship intention. Furthermore, it is observed that innovative behavior has a partially mediating impact on the effect of digital leadership on job performance. Considering the results, this study proves that digital leaders need to adopt innovative behavior so as to ensure performance and intrapreneurship in an organization.

## 1. Introduction

In various countries worldwide, there is an intense contemporary interest in technology as a pioneer of progress and development. Meanwhile, it is understood that many projects are ongoing, which involve integrated uses of technology with the aim of eliminating problems caused by aging populations and deprivation in developed countries. “Made in China 2025”, “UK Industry 2025”, “Industry 4.0”, and “Society 5.0” are some of the projects that come to mind first [1]. These developments have led to the emergence of digital leadership, a novel leadership approach. Digital leaders are expected to transform organizations innovatively and adapt them to contemporary conditions, as well as to increase their performance by managing organizations effectively throughout the process [2]. Research conducted with employees to direct the process in question indicated a high level of belief (77%) that change can be made in organizations through new leaders [3], which was also supported by a report prepared by the European Union stating that there was a need for 40,000 to 50,000 digital leaders every year from 2015 until 2020 [4,5]. Countries with a strategic perspective allocate significant financial resources to digital transformation. Over USD 380 billion is thought to have been allocated for digital transformation in the Asia Pacific Region alone in 2019; this figure increases day by day. It is expected that more than 65% of the Asia Pacific Gross Domestic Product (GDP) will become digitalized due to the USD 1.2 trillion invested in digital transformation between 2020 and 2023 [6].

Excessive workforce needs and considerable financial factors requiring management have fostered an increasing interest in digital leadership studies. Although the positive effect of digital leadership on job performance is known [5,7,8], the mediating role of innovative behavior has not been fully clarified in the literature. Therefore, determining the mediating effect in question constitutes one of the aims of the present study. Moreover, the present study also enquires as to the mediating role of innovative behavior and the effect of digital leadership on intrapreneurship intentions [9,10]. The determined objectives reveal the power of digital leadership in terms of influencing intrapreneurship and job performance; however, they also explain the mediating role of innovative behavior in this effect. In explaining the obtained results, Upper Echelon Theory [11], Social Exchange Theory [12], Social Capital Theory [13], Social Impact Theory [14], Social Information Process Theory [15], Resource-Based View Theory [16], and Dynamic Capability Theory [17] are utilized.

This study bears importance in that it expands the extant literature on digital leadership and innovative behavior. Similarly, an explanation of the impact of digital leadership and innovative behavior on intrapreneurship intention and job performance is expected to contribute considerably to the development of the literature. This study will be helpful for future research because it includes four different variables that are expected to influence each other. Assessing the original contributions, these four variables are actually the issues that have been expected and written about in all studies, but they cannot represent an example adequately. In this respect, it would be appropriate to state that the concept of digital leadership is a very new type of leadership, becoming widespread in Industry 4.0 [2]. The first study on digital leadership was published in 2004, according to Web of Science. In total, 173 articles have been published in the last nineteen years; 139 of these articles have been published in the last five years. Notably, a similar situation is apparent in the Scopus database. As demonstrated from the example given, the present study is expected to contribute to the limited literature on digital leadership through its multivariable structure approach.

The rest of this article is organized into four sections. Firstly, the variables are explained in accordance with a literature review. In the following section, hypotheses regarding the variables are established in accordance with the literature. Information technology employees were selected as the participants for the present study, and this choice was made because digitalization in the IT sector is rapid and intrapreneurship intentions can easily be cultured. Structural equation modeling was utilized as the study method. Findings were analyzed using the Smart-PLS program. Within the scope of the findings, hypotheses were tested, and conclusions were drawn by discussing them in accordance with the literature. According to the obtained results, a partial mediating effect of innovative behavior on the effect of digital leadership on job performance was identified, and a full mediation effect on the effect of digital leadership on intrapreneurship intention was found. Thus, it is possible to state that some of the positive impact created by movement and digital leadership stems from innovative behavior.

## 2. Literature Review

### 2.1. Digital Leadership

Digital leadership first emerged as a leadership approach aiming to follow and implement changes due to technological developments that have arisen with Industry 4.0 [2,8]. A digital leader is expected to be someone who is visionary, sympathetic, agile, a risk-taker, and always open to collaboration [18]. In this regard, digital leaders are expected to create an effective organizational culture by developing social capital. Social Capital Theory, used to eliminate and explain the instability and social problems that arise especially in the industrializing West, appears to be an important tool in terms of explaining digital leadership. The understanding that social capital should be regarded not only as an element of success achieved by the individual work and effort of a person but also as the success of an entire organization contributes to the subject in this respect [19]. This situation, integrated with the understanding of cognitive social capital, is also affected by Social Impact Theory because it creates social impact and gives direction to the relationships within the organization. The stated theories suggest that a digital leader will socially influence the people around him, emphasizing his importance in increasing or decreasing the performance of the organization [20].

Digital leadership refers to the ability of leaders to manage digital-age organizations effectively [21]. In addition to good business skills, digital leaders need to possess good practical knowledge, practical problem-solving ability, and the ability to use and teach digital tools [22] due to the fact that digital leadership aims to understand the effect of technology on business operations and thus manage innovation. Digital leadership involves developing and utilizing technology in order to drive growth and success [23]. Briefly, digital leadership involves adopting and implementing the adaptation of technological developments as a leadership approach with the use of modern technological elements [24]. This behavior of the digital leader can be stated to be within the scope of Resource-Based View Theory in terms of ensuring sustainable competitive advantage [25].

Dynamic Capability Theory states that the value of all resources of the organization, such as human resources, capital, and production ability, is more than the value of the resources alone. A suitable leader is required for the synergy that is expected to be created within the organization [26]. A digital leader has a clear vision in terms of how to use technology so as to improve the organization. In this regard, a strategy is created by the leader to achieve this vision [25]. In order to create a good strategy, digital leaders are expected to have the competence to manage the change process and to have a critical understanding. In accordance with the needs of the organization for change, it is required to take strategic steps regarding new business models, understanding of customer relations, solutions for employees, operational improvements, and financial conditions [18]. In this respect, a digital leader is defined as a person who can show two-way innovative behavior on how to manage the same organization digitally as well as leading the digital transformation of an organization [27].

A digital leader is expected to collaborate with employees and communicate across multiple channels effectively [28]. In accordance with the digital age, leaders must be able to use media tools effectively, make quick decisions, take preventive measures against destructive situations, and possess technical skills [18]. Making the most appropriate decision thanks to big data by maintaining communication with customers uninterruptedly is an important element expected from the digital leader as well [29]. With the help of uninterrupted communication, technological innovations, and innovative behaviors, a digital leader is expected to increase the job performance of the organization [6,29,30,31,32,33].

### 2.2. Intrapreneurship Intention

Entrepreneurship was first used conceptually by Cantillon [34], who mentioned three important elements in shaping the economic structure and defined these elements as economic agents by listing them as capital owners, entrepreneurs, and employees [35]. An entrepreneur, defined by Cantillon as an economic agent, is one who possesses a high tolerance for uncertainty, can display active and innovative behavior, can take the necessary financial risks to develop new projects, and can make commercial commitments [36].

Entrepreneurship requires creating an organization that has not existed before or restructuring an existing organization with a different perspective. In their definition of entrepreneurship, Green and Cohen [37] list the main factors of entrepreneurship as being creative, seizing the opportunity, taking risks, displaying demand for growth, and being profit-oriented. In the context of the economic added value formed, entrepreneurs play an important role in shaping social dynamics and realizing the structural change needed as well as providing employment, welfare, and a demand-oriented supply [38]. From this perspective, it is possible to state that entrepreneurship pioneers digital leadership and innovation in terms of achieving change and spreading innovation.

Entrepreneurship refers to the independent establishment of a new business, and it has also existed as the emergence of new entrepreneurs within the organization, especially since the 1980s. It is known that the method called intrapreneurship, which creates new production models and opportunities with its views and ideas within the organization, has a great effect, especially on production performance [39]. In accordance with the present study, digital leaders are expected to contribute to and develop employees’ intrapreneurship intentions. In this respect, it is predicted that RBV, which is stated to affect digital leadership, will be effective. Obtaining foresight in terms of developing the limited resources at hand and using them as a strategic competitive element constitutes an opinion on the effectiveness of this theory [9,40].

### 2.3. Innovative Behavior

Innovation is stated to be a holistic management process that includes elements such as ideas, technology, manufacturing, and marketing for a new product or production process [41]. Hoecht and Trott [42] describe the concept of innovation as the sum of theoretical concepts, technical invention, and commercial effect. It is defined by Drucker as innovation-oriented activities that are carried out with the aim of developing the organizational activities to be accomplished and the products and services to be produced in line with certain purposes, emphasizing that innovation is a prerequisite for organizations so as to maintain their existence [43]. According to Drucker, organizations that fail to achieve innovation will lag behind changes and will not be able to meet both their organizational and environmental needs, and as a result, they will not be able to maintain their existence and finally will disappear [44].

In general, organizations are expected to achieve meaningful economic value with the help of innovation. In fact, organizations intend to gain benefits such as increasing their profits, reducing their costs, and gaining competitive advantage with innovation. Furthermore, innovation is expected to create some non-material benefits for organizations [45]. Outputs with high organizational importance such as improvement in personal relationships, increase in performance, job satisfaction, personal development, etc., are some of these benefits [46].

Innovation is defined to be a complex process that is dominated by knowledge, requires thought leadership, is associated with the demand for change and transformation, and requires effective management [47], which reveals that the innovation process is related to leadership. In this respect, in the present world where digitalization is the focus of change, it is obvious that the concept of innovation must be advocated by digital leadership [9,48,49]. Innovative behavior is expected to emerge as a result of the incentive environment that is created by the leader. In accordance with Social Impact Theory, the leader’s innovative support is expected to turn into a behavior, which can emerge positively or negatively depending on the leader’s style [5].

### 2.4. Job Performance

Job performance refers to how well an employee fulfills the duties and responsibilities that are assigned to him at the workplace [50]. In other words, the concept of job performance is defined as a measure of how effectively the employee accomplishes his job responsibilities and achieves his goals and objectives [51]. Job performance typically involves performance evaluations made by supervisors or managers, which can influence various organizational outcomes including productivity, job satisfaction, and turnover [5,52,53]. By understanding the factors that have an impact on job performance and using effective performance evaluation methods, organizations can promote high levels of job performance among their employees [54]. In this regard, there are many theories available in the literature. For example, within the scope of Social Exchange Theory [12], it is suggested that positive behaviors exhibited by the leader toward employees will obtain positive feedback from employees within the framework of reciprocity theories. Hence, it is claimed that employees who perceive positive and good behavior of leaders toward themselves are motivated to work harder, which in turn increases the organization’s job performance [55].

Under present conditions, technology and information sharing are used commonly to measure job performance, which is triggered by the ease of obtaining information thanks to various software, especially in current digitalized conditions. Knowledge sharing refers to the exchange of information among employees with the aim of performing tasks in organizations. As a result of information exchange, the job performance of the organization is ensured to increase together with important gains such as the participation of employees in management, effective decision-making, a reduction in information loss, avoiding the repetition of mistakes, and encouraging innovation [56]. Therefore, information sharing has a profound impact on job performance. Within the scope of Social Capital Theory, organizational culture is influenced significantly by coordination, communication channels, and information sharing, which leads to better job performance [57]. From this point of view, it is expected to contribute positively to job performance [5].

## 3. Materials and Methods

### 3.1. Research Model and Hypotheses

#### 3.1.1. Digital Leadership and Innovative Behavior

Digital leadership refers to the ability to use digital technologies effectively in order to manage a team or an organization [58]. Digital leadership can also be defined as an attempt to introduce innovation by encouraging experimentation and risk-taking [24]. The concept of digital leadership should not be perceived merely as the digitalization of an analog clock or an indicator in a system [8]. Digitalized workflows and management philosophy make innovative thinking necessary so as to manage and develop different businesses. Digital leadership includes a structure that can change and renew the existing management style completely. Thus, digital leadership and innovation are important variables that are intertwined and have the possibility to affect each other. When the literature is considered, many studies conducted on digital leadership and innovation can be found [2,22,30,59]. Previous studies indicate that digital leadership has a strong impact on innovative behavior [59] in accordance with Upper Echelon Theory [11]. Based on these, hypothesis H_1_ was established.

**H_1_.** 
*Digital leadership affects innovative behavior in a significant and positive way.*


#### 3.1.2. Digital Leadership and Intrapreneurship Intention

Digital leadership addresses a transformation and development process in businesses. Leaders with entrepreneurial spirit are required, especially in order to find an intrapreneurship intention culture that is sustainable and can develop within the business [10]. In the study, leaders are expected to develop the present organization, encourage employees, and disseminate the in-house entrepreneurship culture rather than leaving the organization and founding their own businesses. With the in-house entrepreneurship culture, many important and positive changes such as supporting research and development activities, creating innovative products or services, capturing opportunities in the sector, creating value for customers, and increasing employee performance take place. In spite of its positive effects, supporting in-house entrepreneurship may be perceived by some leaders as training rival managers or leaders for themselves in the future. For this reason, it is likely that some of the leaders do not support in-house initiatives. However, this situation could also cause employees to become competitors by founding their own companies [9]. Therefore, digital leaders are required to spread the entrepreneurship culture within the organization and keep qualified employees within the organization. RBV Theory, one of the theories on which the present study is based, foresees the best use of employees as a resource as well. From this perspective, it is anticipated that digital leaders, as managers with an entrepreneurial vision, support in-house entrepreneurship and provide sustainable competitive advantages by creating core capabilities of the organization [16]. In this respect, hypothesis H_2_ was established.

**H_2_.** 
*Digital leadership affects intrapreneurship intention significantly and positively.*


#### 3.1.3. Digital Leadership and Job Performance

Digital leadership can easily improve communication within a business with the use of more efficient, transparent, and accessible technologies [8]. When leaders use digital tools while communicating with their teams, they can enable easier access to members and pave the way for a more conciliatory environment among stakeholders regarding the management process [29]. In this respect, the difference between communication via fax, where pages are scanned one by one, and e-mail, where thousands of pages of data are sent with a single click, reflects the positive difference in terms of job performance. Additionally, it ensures that tasks are completed on time with minimum errors, which results in an increase in job performance. Digital leaders make more functional decisions because they can access real-time data. It can be stated that their decisions are more consistent and healthy and increase job performance because they are made based on data [7]. The transformation that is experienced via digital leadership can lead to new ideas and approaches that can improve job performance and accelerate growth. Based on the opinions stated above, hypothesis H_3_ was established.

**H_3_.** 
*Digital leadership affects job performance in a significant and positive way.*


#### 3.1.4. The Relationship between Innovative Behavior, Intrapreneurship Intention, and Job Performance

Schumpeter [60] considers entrepreneurship to be a concept that is inseparably associated with innovation and defines it as the process of creating new products and organizations while describing the entrepreneur as the bearer of innovations that are necessary for economic development, development, and change. This innovation role that is attributed to the entrepreneur by Schumpeter was also supported by Baumol [61], and it was suggested that investments in innovation and technology transfers could only be accomplished through entrepreneurial practices [41]. It is possible to divide entrepreneurship practices into two, which involve an initiative independent of the organization and internal initiatives within the organization. Intrapreneurship has a great impact on the organization in terms of introducing innovative products and processes. Whether the organization is large or small does not diminish intrapreneurship [62]. As an enterprise, it is known that internal initiatives contribute to innovation and to the performance of the business in terms of introducing new products and services within the organization, caring about the opinions of employees, and meeting the demands of customers [39,63,64,65]. In this study, it is expected that innovation will be encouraged within the organization and contribute to performance by turning into behavior in the context of intrapreneurship. In accordance with the stated opinions, hypothesis H_4_ was established.

**H_4_.** 
*Innovative behavior affects intrapreneurship intention significantly and positively.*


The concepts of intrapreneurship and innovation have enabled job performance to increase while expanding the range of innovative production, especially with destructive creation [60]. The previous research provides much evidence that designing new products and processes increases job performance [66,67]. Business processes are expected to become easier and show development, especially with digital transformation. As a phenomenon, this expectation has emerged with Industry 4.0, with which great potential is exhibited in creating 30% faster and 25% more efficient production in industry [68]. Creating new products and processes that give priority to efficiency and profitability increases the effect of innovation on job performance. Bearing these statements in mind, hypothesis H_5_ was established.

**H_5_.** 
*Innovative behavior affects job performance significantly and positively.*


#### 3.1.5. The Mediating Role of Innovative Behavior

Digital leadership, which includes using technology and digital tools in order to communicate, collaborate, take initiative, and innovate, is also described as a form of transformational leadership [29]. By leveraging digital technologies, leaders can create a better-equipped, more connected, and engaged workforce that can meet the demands of a rapidly changing digital world. In the digital age, in which technology and digital tools play an increasingly important role in how jobs are performed, digital leaders who are competent at using digital technologies are more likely to be effective in leading their teams to success [69]. In this respect, it should be stated that the ability to take initiative comes to the fore. Despite all the suitable environments, the most important element for the leader will be to make an attempt, that is, to take the initiative. Timing is also an important factor in initiative [9]. The initiatives taken by the leader are expected to increase the innovative behavior of employees and create tendencies toward intrapreneurship. It is thought that digital leaders, who are stated to have a transformational leadership quality, will create an innovative organizational culture along with internal initiatives [18,29,59]. It is possible to reveal the intentions and behaviors of employees in a planned manner with the appropriate environment and the encouragement of employees by leaders. According to the Theory of Planned Behavior [70], employees will possess a great desire to exhibit the desired behavior if they have the necessary resources and opportunities. As a result, taking initiatives within the organization and introducing new processes will strengthen the leader’s position and will increase the promotions and salaries of his employees within the organization [71]. In accordance with the opinions stated above, hypothesis H6 was established.

**H_6_.** 
*Innovative behavior mediates the effect of digital leadership on intrapreneurship intention in a meaningful and positive way.*


Innovation can improve job performance by introducing new tools and processes that have the potential to increase efficiency, productivity, and quality [53]. Digital leaders who can trigger and support innovation can create a culture that gives value to creativity and encourages employees to think distinctively [9], which in turn leads to an increase in job satisfaction, motivation, and commitment, all of which will lead to better job performance. The Resource-Based View suggests that a firm’s resources and capabilities can make great contributions to competitive advantage [16]. Digital leadership can be regarded as a resource or capability that could be utilized to lead innovation and improve job performance. Digital leaders can create new processes and products that can potentially increase job performance using digital tools and technologies [5]. Based on those opinions, hypothesis H_7_ was established.

**H_7_.** 
*Innovative behavior significantly and positively mediates the effect of digital leadership on job performance.*


The research model created with the aim of understanding the models and hypotheses better is presented in Figure 1.

When the model is considered, it is realized that some variables have been examined many times beforehand by other researchers. Highlighting digital leadership, which is a relatively new concept, in the present study reveals and proves its originality.

There are limited studies in the literature that show how digital leadership affects innovative behavior [18,29]. The H_1_ part of the created model is similar to the study on digital leadership and innovative work behavior conducted by Erhan et al. [29] based on Upper Echelon Theory. In Erhan et al.’s study, two variables were examined, and it was stated that theoretically, these variables would affect work performance positively. It can be claimed that the present study expanded the principles stated by Erhan et al. [29] within the framework of H_3_ and H_7_, thus making both theoretical and practical contributions to the literature.

When studies measuring the impact of digital leadership on performance are taken into consideration, it is realized that the relationship is positive and significant. In this respect, the study conducted by Azzam et al. [72], based on Resource-Based View Theory [16] and Dynamic Capability Theory [17], comes to the fore. In that prominent study, there is a structure that measures the mediating role of digital leadership and entrepreneurial orientation in the effect of dynamic capabilities on competitive performance. The concept of innovation is frequently discussed by Azzam et al. in their study. However, its relationship with other variables was only attempted to be explained theoretically. Therefore, it is thought that the present study will make both theoretical and practical contributions to the literature in supporting Azzam et al.’s study by including innovative behavior as well.

### 3.2. Methodology

This study was carried out with employees in the IT sector in Istanbul. Before obtaining ethics permission, 62 businesses with employee numbers ranging from 10 to 280 were visited. By giving information about the questions in the survey form to the businesses that were visited, managers and business owners were informed about the aim of this study and its possible consequences. This information process was performed as preliminary field research to determine the research areas, so it was not possible to have the interviewed people fill out any survey forms or include them in the analysis. The data were collected with the use of the convenience sampling method. With the aim of collecting data, businesses in the Sisli and Besiktas districts in Istanbul, where the IT sector is more common, were preferred. There are many authorized services, wholesalers, and service providers in the IT sector in the region in question. In order to collect data, permission was received from the Istanbul Arel University Ethics Committee, dated 8 September 2023, and numbered 2023/18-7. After obtaining the necessary permission, a method that utilized a printed survey form and a digital survey form was used to collect the data between 8 September 2023 and 11 September 2023. The survey was chosen due to the fact that it provided generalizable, valid, and reliable findings. The present situation was measured cross-sectionally with the help of a quantitative approach. Following the distribution of the survey, 390 people were reached, which was enough to represent the sample [73]. Data belonging to these 390 people were obtained from 21 different businesses. The information provided before obtaining ethical permission was effective in terms of collecting the data for the present study in a short time. Thanks to the information activity, it became easier to meet managers and, in this way, businesses that could support the study were identified. This is an obligatory step in terms of filling out the section related to the research field that is specified on the ethics committee form.

Questions related to age, gender, marital status, education level, and experience were used in the survey form so as to categorize the participants. Demographic variables are an important element for showing whether the participants are distributed homogeneously or whether only one group is represented [74]. In Turkey, employees are expected to be predominantly male. In this regard, demographic variables are needed to determine whether only male individuals’ opinions are included in this study. In addition, sharing this issue is an important element in order for the analysis to be repeated by other researchers. For example, if the same questions are asked to individuals over the age of 50 and those under the age of 20, it is not always possible to obtain the same answers. For these reasons, sharing demographic variables comes to the fore. Demographic variables also provide information about the sector of employees [75].

In order to test the model, scales previously developed by other researchers and applied in Turkey were utilized. All scales were used in a 5-point Likert form. In order to measure digital leadership, a one-dimensional 6-question scale, developed by Zeike et al. [69] and translated into Turkish by Oktaysoy et al. [76], was used. The Cronbach’s Alpha coefficient of the digital leadership scale was determined to be 0.870 by Zeike et al. in 2019, and questions such as “I think using digital tools are fun” were included. On the other hand, to measure intrapreneurship intention, a one-dimensional 6-question scale, developed by Liñán and Chen [77] and translated into Turkish by Basim and Sesen [78], was used. The Cronbach’s Alpha coefficient of the entrepreneurial intention scale was determined to be 0.943 by Liñán and Chen in 2009, and questions such as “I am ready to do anything to become an entrepreneur” were included. For innovative behavior, a one-dimensional 9-question scale, developed by Janssen [79] and translated into Turkish by Onhon [80], was utilized. Cronbach’s Alpha coefficient of the innovative behavior scale was determined to be 0.950 by Janssen in 2000, and questions such as “Creating new ideas for difficult issues” were included. To measure job performance, a one-dimensional 4-question scale, developed by Sigler and Pearson [81] and translated into Turkish by Col [82], was used. The Cronbach’s Alpha coefficient of the job performance scale was determined to be 0.830 by Sigler and Pearson in 2000, and questions such as “I complete my tasks on time” were included.

With this study, the mediating role of innovative behavior on the effect of digital leadership on intrapreneurship intention and job performance was planned to be measured. Statistical mediation analysis is utilized to investigate how an independent variable (X) affects an independent variable (Y) through a mediating variable (M). For example, in one part of this study, the mediating effect of innovative behavior (M) on the effect of digital leadership (X) on intrapreneurship intention (Y) is investigated. As a result of this research, there may or may not be a mediating effect. If a mediating effect is detected, this result is in the form of a partial or full mediation. The main aim of the mediation analysis is to determine the reason for the interaction between the independent variable and the outcome variable. In this regard, if the mediating variable can explain the entire interaction, it is called a full mediator, whereas it is called a partial mediator if it can explain only a part of it. Structural equation modeling along with regression analysis can be utilized in order to measure mediation analysis. In order to identify the mediating role, there must first be a relationship between the variables. Many different methods can be used for mediation testing such as the model proposed by Baron and Kenny [83], the Sobel test, and the Goodman test [84]. As one of the newly used techniques, the bootstrap technique is used to overcome the reliability problems that are encountered with the Sobel test. The technique in question creates a new and different data set by producing new observations in the specified number of data sets. Statistical calculations are made according to this new data set, and errors that may arise from bias and skewness are corrected in this way [85]. In the analysis performed, the sampling number was determined to be 5000. Structural equation modeling was utilized to test the hypotheses. The least-squares method (SEM), which is the most suitable method for testing, was preferred. In this respect, the Smart-PLS program was utilized, which enables confirmatory factor analysis, validity, reliability, internal consistency, normality distribution, and structural equation modeling to be practiced easily. Smart-PLS was preferred especially because it allows the testing of complex structures, multiple variables, and big data simultaneously [86]. There is a motivation for choosing this method due to the fact that it contains more than one variable both visually and in terms of the ease of analysis for complex models. For instance, in the Smart-PLS program, incorrect and missing expressions in variables and normality distribution issues are reflected on the screen as soon as the data set is loaded. Analysis using SPSS requires effort for all of these issues pointed out above. To perform the confirmatory factor test, variables are created in the path with drag and drop logic, and many issues such as factor load values, validity, reliability, and multicollinearity are tested using a single button with the Calculate (PLS Algorithm) screen at the top of the program. In the mediation analysis, the mediation test is easily performed using the Calculate (Bootstrapping) screen at the top of the program. Researchers are provided with great convenience because all calculations and visualizations are enabled by the program as printouts and Excel files. When compared with SPSS, the program gives exactly the same Cronbach’s Alpha values and allows for an analysis that could be performed using SPSS and AMOS to be easily carried out [86].

## 4. Results

In the present study, 390 people were reached, and the information about the participants is displayed in Table 1. It was observed that the participants were dominantly male (67.20%). It can be stated that this result was achieved in accordance with the present conditions of Turkey, although the IT sector is a sector where more women are employed as a result of not requiring heavy muscle strength. It was also observed that the number of married participants (61.80%) was higher than that of single participants. It was found that the participants of the study were mostly between the ages of 36 and 45 (36.40%), had a bachelor’s degree (54.60%), and had 6 years or more of experience (82.60%).

There are various methods utilized to measure the validity, reliability, and internal consistency of structurally used scales, the most well-known of which are Cronbach’s Alpha, Composite Reliability (CR), and Average Variance Extracted (AVE). Cronbach’s Alpha is frequently preferred for measuring combined reliability and internal consistency values. However, the use of the CR value comes to the fore to fill in this gap due to the fact that it is known that the probability of Cronbach’s Alpha obtaining better values increases as the number of samples and the number of questions increases. For convergent validity, the CR value is expected to be greater than the AVE value [87]. Information regarding the analyses performed is presented in Table 2.

According to the literature, Cronbach’s Alpha, rho_A, and CR coefficients are expected to be above 0.70, while AVE value is estimated to be above 0.50, i.e., CR > AVE [86,87]. Moreover, factor load values are required to be above 0.50 for each variable individually. As the values for all variables are within the desired criteria, it can be stated that the scales do not have any problems in terms of validity and reliability. The kurtosis and skewness values are between +1.96 and −1.96, which indicate that the sample is normally distributed [73]. When the values obtained as a result of the analysis are evaluated, the scales and the sample can be stated to be suitable for the research.

Fronell and Larcker suggest that AVE values should have a value higher than the correlation of the variables [87]. In this way, it is accepted that discriminant validity between the scales is ensured. Another method, the Heterotrait–Monotrait Ratio, suggests that having scales with ratios below 0.85 ensures discriminant validity [88]. The results of this study reveal that it has discriminant validity. The results are presented in Table 3.

The goodness and meaningfulness of the model that provides discriminant validity, reliability, and validity were measured. In this regard, the goodness of fit values of the model are presented in Table 4. In terms of values, the Standardized Root Mean Square (SRMR) value is expected to be below 0.80 and the Normed Fit Index (NFI) value is expected to be above 0.80 when the number of samples and the number of questions in the survey are taken into consideration [89].

Since the obtained values met the necessary conditions for the application of the model, the model and hypotheses were tested using structural equation modeling. The model was performed in accordance with the literature, using the bootstrap method and selecting a sample size of 5.000. In order to provide a better understanding of the created model, the current situation is displayed in Figure 2.

The results of the analysis indicated that all of the hypotheses were supported by values of *p* and *t*. The significant relationship between digital leadership and intrapreneurship intention became meaningless with the full mediating effect of innovative behavior, especially within the scope of H_2_. The hypothesis results and findings regarding the model are presented in Table 5.

Digital leadership was found to have a significant effect on innovative behavior and, therefore, H_1_ was supported. It was observed that the findings obtained and the research on the relationship between digital leadership and innovation were compatible [2,22,90]. Meanwhile, the situation in question is also explained by Upper Echelon Theory. According to the theory, the leader’s past experience and abilities will have an impact on the decisions the leader will make. Bearing this in mind, it is considered that the decisions made by the digital leader will have a great impact mainly because of the aspect of innovation [5,40,59].

It was observed that digital leadership had a significant effect on intrapreneurship, and in the model, this effect decreased as a result of the mediating effect. When it was considered outside the mediating effect, it was seen that H_2_ was supported. When the result obtained was compared to the findings in the literature, it should be stated that the literature was created conceptually with a small number of studies [9,10], and the finding was below expectations. However, in terms of contributions to the literature, it must be said that the finding in question has a small effect in terms of filling a gap in the present literature.

It was found that digital leadership had a significant effect on job performance and, therefore, H_3_ was supported. In this respect, the findings obtained and the research on the relationship between digital leadership and job performance were realized to be compatible [5,91]. Efficiency and productivity are thought to increase as a result of a more effective management approach with digital tools. Within the scope of Social Impact Theory [14], the behavior of digital leaders is expected to affect their employees and increase job performance [5].

Furthermore, it was observed that innovative behavior had a significant effect on intrapreneurship intention, and thus, H_4_ was supported. When the results obtained are compared to the literature, the values can be claimed to be sufficient [41,43,92].

Innovative behavior was observed to have a significant effect on job performance and, therefore, H_5_ was supported. When the results obtained are compared to the literature, the values can be claimed to be sufficient [67,93,94].

Furthermore, innovative behavior was realized to have a significant mediating impact on the effect of digital leadership on intrapreneurship intention and, therefore, H_6_ was supported. The result shows that innovative behavior is the latent variable underlying the digital leader’s realization of his entrepreneurial intention within the organization, which seems to be compatible with the literature [9,10]. One of the first theories to explain the deep-rooted relationship between innovation and entrepreneurship is Schumpeter’s theory of creative destruction [60]. According to this theory, it becomes obvious that organizations introduce technology and innovations and create changes in economic order. When the idea that organizations need innovation and initiatives in order to make more profit and survive is considered, the result is supported by the theory in question. In this respect, it is also realized that the leader must act in a way that supports destructive creation by using the resources and information in his hand in the best way. Wang et al. [40] explains the result using Social Information Process Theory (SIP) [15] and Resource-Based View Theory (RBV) [16]. Technological developments have brought about the necessity of increasing the qualifications of employees. In this regard, RBV guides employees in accordance with the leader’s tendency, provides opportunities for their development, and encourages them to take innovative initiatives. Many elements such as big data, sensors, e-mails, and cyber–physical systems have taken their place in the understanding of leadership that experiences digital transformation. For this reason, information must be processed, classified, and used socially in order to realize innovative initiatives. It can be stated that SIP provides great help to digital leaders while processing information.

The results showed that innovative behavior had a significant mediating impact on the effect of digital leadership on job performance and thus, H_7_ was supported, which reveals that the result obtained is compatible with the literature. The result shows that innovative behavior is the latent variable underlying the digital leader’s increase in job performance within the organization. The fact that only a part of the change in performance is explained by innovative behavior may be related to the fact that employees are influenced by other variables due to their humanistic aspects. Especially with the influence of the atmosphere in the organization, it is possible to find other opportunities such as increases in salaries, promotions, gifts, holidays, and remote working opportunities that will motivate employees more. From this perspective, it can be realized that job performance is affected by more than one variable at the same time and innovative behavior is among these variables. It can also be stated that the leader has a higher weight among all these variables due to his guiding and permissive role [67,93,94]. When considered within the scope of Dynamic Capability Theory (DCT) [17], this points to the adaptation process itself in terms of management, equipment, and staff in the organization against the changes and renewals occurring in the structure of the organization. According to DCT, experiencing an increase in job performance as a result of the fact that the change process beginning with digital leadership is accepted and implemented by employees is an expected result [2].

One of the methods recommended to determine the accuracy of analyses related to structural models is the Q^2^ value. The Q^2^ value should be above zero. The other value that should be explained is R^2^, where the R^2^ value is regarded as a measure of the predictive power of structural equation models [87]. Information regarding the analysis conducted in this regard is illustrated in Table 6.

The findings that were obtained so far will be discussed in the conclusion section and presented to the reader by comparing the results with the literature.

## 5. Discussion and Conclusions

The present study aimed to explain the mediating role of innovative behavior on the effect of the concept of digital leadership on other variables including job performance and intrapreneurship intention as well as to contribute to the literature. These concepts were examined one by one by many researchers in the previous literature. However, in this study, we attempt to explain the interaction in question as a whole using more than one theory, in contrast to other studies.

The impact of digital leadership on innovative behavior was found to be as expected. It is possible to explain this development using the Upper Echelon Theory [11], which states that organizational results are influenced by the characteristics of the organization’s top managers. For instance, it is argued by this theory that the characteristics of the top manager, such as beliefs, values, attitude, and professional competence, have a great effect on decision-making within the organization [95]. The result is explained by the fact that one of the measured variables is related to innovative behavior, which indicates the characteristics of top management and that it is not possible for a manager who adopts innovation to avoid technological transformation [22].

One of the most important responsibilities of digital leaders in the organization is to lead a sustainable change in the structure of the organization by ensuring digital transformation. In order to achieve this change, the leader must develop an entrepreneurial spirit [9]. In the literature, it is clear that the relationship between entrepreneurship and digital leadership is explained in a limited number of studies [10]. From this point of view, the present study is expected to contribute to the development of the literature, even if it is only a small contribution. When the situation resulting from the analysis was evaluated, it was observed that the effect of digital leadership on intrapreneurship intention was below the expected level. In particular, it is possible for Social Impact Theory to be effective based on the results obtained. Social Impact Theory suggests that an individual’s beliefs, behaviors, and attitudes emerge as they are influenced by the people around them [96]. In this respect, the result obtained can be claimed to be a reflection of the economic uncertainties that Turkey has been experiencing in the last few years.

The effect of digital leadership on job performance was observed to be at the desired level. Within the scope of Social Exchange Theory, it is expected that a digital leader’s transformational, innovative, communicative, and supportive attitude will motivate his employees to exhibit higher performance [5,91]. Similarly, the effect in question can possibly be explained using Social Capital Theory [13], Social Impact Theory [14], Social Information Process Theory (SIP) [15], Resource-Based View Theory [16], and Dynamic Capability Theory as well [17].

Entrepreneurship is an important variable that reveals destructive formation by presenting an inseparable integrity of innovation [60]. Many previous studies have proven that there is a strong relationship between innovative behavior and entrepreneurship [41,43,92], which is repeated by the situation resulting from the analysis in this study. 

As a result of technological developments and thus the emergence of new production methods and products, innovative behavior is expected to have a positive and significant effect on job performance [66,67]. The results of this study conducted in accordance with this expectation are similar to the results in the literature. It seems possible that the result obtained within the scope of Social Information Process Theory (SIP) [15] can be associated with the development of employees’ attitudes and behaviors in light of the information that they acquired by examining the environment. It can be said that the expected belief in terms of increasing efficiency and effectiveness as a result of innovation is a kind of manifestation of the result obtained.

Digital leaders are observed to encourage innovative behavior and enable the formation of new initiatives, which can be not only within the business but also in the form of establishing another business outside the business [9]. Within the scope of the present study, only the entrepreneurship factor within the organization was tested, and it was found that innovative behavior had a full mediating role in the effect of digital leadership on intrapreneurship intention, which is based on the fact that motivating employees in order to acquire unique and inimitable features within the scope of Resource-Based View Theory [16] paves the way for innovative behaviors. Especially in the IT sector, where competition is intense, a leader’s perspective strategies and perspective are important variables in terms of the survival of the organization, the creation of new initiatives, and insurance of customer satisfaction. Flexible working hours, rewards, promotions, and various incentives for the generation of new ideas in accordance with the developed strategies and the development of the social environment are frequently encountered in the IT sector [97]. It is possible to talk about the social impact as a result of creating and developing a suitable environment, which paves the way for the realization of new initiatives. Social impact helps employees come up with more innovative ideas and create internal initiatives. The possibility that the mediating role detected between the variables may be a result of employees’ social interaction with the workplace can be explained using Social Impact Theory [14,96]. The result obtained reveals that innovative behavior has a great effect on the development and progress of intrapreneurship intention and more than one theory plays a mediating role in explaining the relationship between digital leadership and intrapreneurship [9].

Innovative behavior was observed to have a partial mediating effect on the effect of digital leadership on job performance. A digital leader restructures the organization by carrying out an innovative transformation and therefore increases the job performance of the organization. When the result of this study is considered within the scope of Dynamic Capability Theory, it reveals that the organization provides an increase in performance in the process of adapting itself in terms of management, equipment, and staff against the changes and renewals occurring in the structure of the organization [2]. In addition, it was found that innovative behavior had a partial role in the increase. Although there are various classifications regarding the concept of innovation, it seems possible to associate the result, especially with incremental innovation, which suggests increasing the performance of a process and product by adding various features and innovations [94]. The cumulative development of the mobile phones that we use today over the last thirty years is a good example of this innovation. From this perspective, leaders’ behavior toward making the organization more productive rather than a destructive creation process reminds us of the similarity between digital leadership and transformational leadership [29].

When the results are considered, the present study proves that a digital leader needs innovative behavior in order to ensure performance in the organization and progress with internal initiatives. On the effect of digital leadership on intrapreneurship intention, a full mediating role of innovative behavior is observed within the scope of Schumpeter’s theory of creative destruction [60]. However, this effect progresses a little more cautiously and slowly in performance. Despite major changes in organizations, the rate of change in performance generally remains limited. For example, it was stated in the previous sections that the positive difference to be achieved through Industry 4.0 is expected to be 30%. Based on this, the present study is expected to contribute to the literature both practically and theoretically.

The fact that this study collected data using a convenience sampling method with 390 people in Istanbul is an aspect that is open to criticism. However, Istanbul was preferred because it is the largest city in Turkey, and one-fifth of Turkey’s population is located in this city alone. The faster advancement of digital transformation and leadership in the IT sector compared with the manufacturing and service sectors [6] was also considered in the selection of this sector. The two issues mentioned before constitute the limitations of the present study. For future studies, testing digital leadership in different sectors will help the subject matter gain prevalence and be understood better. It is thought that it will be useful to examine digital leadership in terms of unemployment, social development, and the United Nations Sustainable Development Goals.

## Figures and Tables

**Figure 1 behavsci-13-00874-f001:**
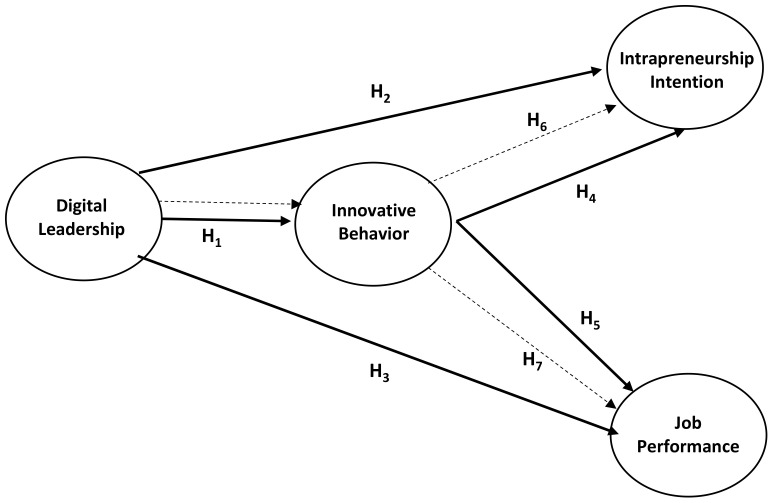
Research model.

**Figure 2 behavsci-13-00874-f002:**
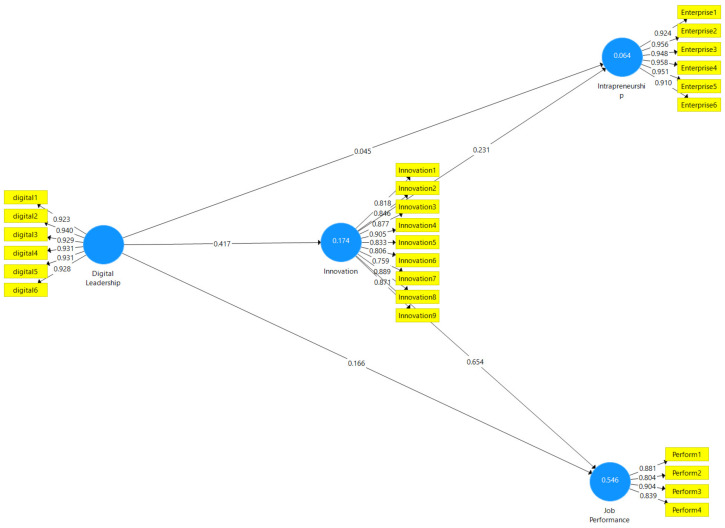
Path diagram of the model.

**Table 1 behavsci-13-00874-t001:** Demographic findings.

Demographic	Items	*n*	%
Gender	Female	128	32.80
	Male	262	67.20
Marital Status	Married	241	61.80
	Single	149	38.20
Age	Between the ages of 18 and 25	63	16.20
	Between the ages of 26 and 35	126	32.30
	Between the ages of 36 and 45	142	36.40
	Between the ages of 46 and 55	59	15.10
Education	Associate degree	154	39.50
	Bachelor’s degree	213	54.60
	Postgraduate degree	23	5.90
Experience	Up to 5 years	68	17.40
	Between the years of 6 and 10	124	31.80
	Between the years of 11 and 15	102	26.20
	Between the years of 16 and 20	84	21.50
	21 years and over	12	3.10

**Table 2 behavsci-13-00874-t002:** Factor loading values, reliability, and validity.

Items	Factor Loading	Mean	Standard Deviation	Kurtosis	Skewness
Digital Leadership Cronbach’s Alpha = 0.969, rho_A = 0.971, CR = 0.975, AVE = 0.865					
Digital 1	0.923	3.838	0.901	1.008	−0.963
Digital 2	0.940	3.779	0.899	0.801	−0.870
Digital 3	0.929	3.785	0.880	1.018	−0.994
Digital 4	0.931	3.651	0.923	0.420	−0.861
Digital 5	0.931	3.746	0.953	0.982	−1.099
Digital 6	0.928	3.638	0.934	0.313	−0.720
Intrapreneurship Intention Cronbach’s Alpha = 0.974, rho_A = 0.977 CR = 0.979, AVE = 0.886					
Enterprise 1	0.924	3.500	1.152	−0.235	−0.738
Enterprise 2	0.956	3.372	1.180	−0.595	−0.537
Enterprise 3	0.948	3.400	1.172	−0.568	−0.617
Enterprise 4	0.958	3.341	1.163	−0.678	−0.534
Enterprise 5	0.951	3.408	1.168	−0.551	−0.595
Enterprise 6	0.910	3.318	1.153	−0.680	−0.412
Innovative Behavior Cronbach’s Alpha = 0.950, rho_A = 0.952, CR = 0.958, AVE = 0.715					
Innovation 1	0.818	3.926	0.812	1.182	−0.845
Innovation 2	0.846	3.705	1.051	0.008	−0.810
Innovation 3	0.877	3.856	0.835	0.869	−0.812
Innovation 4	0.905	3.738	0.913	0.849	−0.837
Innovation 5	0.833	3.797	0.858	0.472	−0.699
Innovation 6	0.806	3.300	1.047	−0.690	−0.113
Innovation 7	0.759	3.564	0.974	0.499	−0.691
Innovation 8	0.889	3.626	0.955	0.754	−0.735
Innovation 9	0.871	3.679	1.016	−0.332	−0.590
Job Performance Cronbach’s Alpha = 0.806, rho_A = 0.847, CR = 0.869, AVE = 0.623					
Perform 1	0.881	3.649	0.872	0.020	−0.509
Perform 2	0.804	3.590	0.842	1.272	−0.919
Perform 3	0.904	3.569	0.859	0.092	−0.533
Perform 4	0.839	3.718	0.756	0.923	−0.551
CR = Composite Reliability, AVE = Average Variance Extracted					

**Table 3 behavsci-13-00874-t003:** Discriminant validity.

Fornell–Larcker Criterion					Heterotrait–Monotrait Ratio (HTMT)			
	1	2	3	4	1	2	3	4
Digital leadership	0.930							
Intrapreneurship intention	0.141	0.941			0.141			
Innovative behavior	0.417	0.250	0.846		0.432	0.256		
Job performance	0.439	0.262	0.723	0.858	0.471	0.280	0.786	

**Table 4 behavsci-13-00874-t004:** Values of the model.

Model Fit		
	Saturated Model	Estimated Model
SRMR	0.050	0.052
d_ULS	0.799	0.885
d_G	0.642	0.645
Chi-square	1.376.881	1.383.239
NFI	0.882	0.881

**Table 5 behavsci-13-00874-t005:** Structural equation model and hypothesis test results.

Path	Estimate	Standard Deviation	*t*-Value	*p*	Hypothesis
Digital leadership -> innovative behavior	0.417	0.059	7.086	0.000	H_1_ accepted
Digital leadership -> intrapreneurship intention	0.045	0.065	0.677	0.498	H_2_ accepted *
Digital leadership -> job performance	0.166	0.041	3.991	0.000	H_3_ accepted
Innovative behavior -> intrapreneurship intention	0.231	0.058	4.009	0.000	H_4_ accepted
Innovation -> job performance	0.654	0.038	17.068	0.000	H_5_ accepted
Digital leadership -> innovative behavior -> intrapreneurship intention	0.097	0.030	3.192	0.001	H_6_ accepted (Complete)
Digital leadership -> innovative behavior -> job performance	0.273	0.043	6.346	0.000	H_7_ accepted (partial)

* Due to the mediating effect, the significant relationship with β = 0.141, *t* = 2.985, and *p* = 0.003 became insignificant.

**Table 6 behavsci-13-00874-t006:** R^2^ and Q^2^ values.

Latent Variable	R^2^	R^2^ Adj.	Q^2^
Intrapreneurship intention	0.064	0.059	0.055
Innovative behavior	0.174	0.172	0.119
Job performance	0.546	0.543	0.392

## Data Availability

The data that support the findings of this study are available from the corresponding author upon reasonable request.

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
