# Peer review of "The Mediating Role of Innovative Behavior on the Effect of Digital Leadership on Intrapreneurship Intention and Job Performance"

_behavsci, 2023, doi:10.3390/bs13100874_

Round 1
Reviewer 1 Report
I thank the authors for the study and the editor for the opportunity to review it.
It is a well-conceptualized and methodologically well-carried-out research work.
From my perspective, it presents two crucial points for improvement. The abstract of the work needs more description of the implications and is therefore not very appealing to read. The other point of progress lies in discussing the work; greater argumentation and debate about the results would be desirable, and the theoretical and practical implications of the study findings should be more substantiated and reflected.
Author Response
Dear Reviewer,
First of all, I would like to express my gratitude for your precious comments and suggestions to improve the article. I took your suggestions into consideration and made the following changes in the article;
- The abstract of the work needs more description of the implications and is therefore not very appealing to read.
The changes your requested have been made in the abstract section. Corrections are highlighted in Red font in the text. Line 29-30. Other changes have been marked in the text in line with the opinions of the other three reviewers.
- The other point of progress lies in discussing the work; greater argumentation and debate about the results would be desirable, and the theoretical and practical implications of the study findings should be more substantiated and reflected.
The changes your requested have been made in the discussion section. Corrections are highlighted in Red font in the text. Line 609-622 and Line 632-650.
Thank you for your great support and attention.
Sincerely,
Reviewer 2 Report
Thank you for allowing me to review "The Mediating Role of Innovative Behavior in the Effect of Digital Leadership on Entrepreneurial Intention and Job Performance."
This study is fascinating, given the ever-growing technological innovation. I have some advice for researchers regarding this study.
Literature Review
- Some variables are structured differently from the study's title and the research model's variables. The consideration of this will also have to be revised. For example, Entrepreneurship, Innovation, etc
Line 346 to 404
- It makes no logical sense to compare β values from previous studies because the sample samples are different.
5. 5. Discussion and Conclusions
- Discussion, theoretical implications, and practical implications should be described separately.
- In conclusion, the importance of this study should be summarized and concluded.
Author Response
Dear Reviewer,
First of all, I would like to express my gratitude for your precious comments and suggestions to improve the article. I took your suggestions into consideration and made the following changes in the article;
- Some variables are structured differently from the study's title and the research model's variables. The consideration of this will also have to be revised. For example, Entrepreneurship, Innovation, etc. (Line 346 to 404)
The changes your requested have been made in the text. Corrections are highlighted in Red font in the text. Other changes have been marked in the text in line with the opinions of the other three reviewers.
- It makes no logical sense to compare β values from previous studies because the sample samples are different.
The changes your requested have been made in the text. Line 494-535.
- Discussion, theoretical implications, and practical implications should be described separately. In conclusion, the importance of this study should be summarized and concluded.
The changes your requested have been made in the discussion section. Corrections are highlighted in Red font in the text. Line 609-622 and Line 632-650.
Thank you for your great support and attention.
Sincerely,
Reviewer 3 Report
Dear Authors,
The reviewed manuscript aims to study the mediating effect of innovative behaviour within the influence of digital leadership upon job performance and entrepreneurial intention, testing the research model on a sample of 390 people working in the Turkish IT sector in Istanbul. Some significant improvements regarding the paper are proposed and needed to be considered:
C1: Concerning the Introductory part, it should be extended with the originality elements and the structure of the paper.
C2: One shortcoming of the reviewed manuscript concerns the theoretical background, which should be extended and correlated the extant literature. On one hand, a deeper analysis of the literature with a more critical approach should be included, on the other hand, the authors should discuss the mentioned theories (from the Introduction) in this section, not just later within the Discussions. Furthermore, the authors should discuss and clarify if their research regards entrepreneurial intentions or intrapreneurial intentions.
C3: Regarding section 3 (Materials and Methods), hypothesis H4 and H6 are not sufficiently argued.
Moreover, within their argument, the Authors should pay attention if they regard intrapreneurial or entrepreneurial intention, considering that respondents are from existing IT companies.
After presenting the research model (Figure 1), the Authors should present the originality elements of their study compared to some other relevant studies.
Concerning the methodological part, the authors should present and argue the used statistical techniques, especially the one to test mediation should be detailed (considering the aim of the study presented witin the abstract), also why is bootstrapping regarded. Eventually some similar studies using these techniques should be referenced.
C4: Concerning the empirical part, in my opinion, betas from other studies should not be invoked, because samples, research contexts, models perhaps are not entirely comparable. Moreover, the mediations should be further tested, discussed, not just stating accepted completely or partially.
C5: The Conclusions part should be extended, by adding managerial and policy implications, and by extending the research limitations and future research directions.
C6: Other observations:
- if the present study considers no big data, then maybe a motivation to apply Smart-PLS should not refer to this advantage (line 288);
- Authors should review the Acknowledgement part.
I hope the above observations will contribute to the improvement of the reviewed manuscript.
Best regards,
The Reviewer
Some sentences should be rephrased: e.g. „Gross Domestic Product (GDP) is expected to be digitalized” (lines 48-49); “the goodness of fit values of the model is presented” (lines 326-327), etc.
Author Response
Dear Reviewer,
First of all, I would like to express my gratitude for your precious comments and suggestions to improve the article. I took your suggestions into consideration and made the following changes in the article;
- Concerning the Introductory part, it should be extended with the originality elements and the structure of the paper.
The changes your requested have been made in the introduction section. Corrections are highlighted in Red font in the text. Other changes have been marked in the text in line with the opinions of the other three reviewers. Line 66-93
- One shortcoming of the reviewed manuscript concerns the theoretical background, which should be extended and correlated the extant literature. On one hand, a deeper analysis of the literature with a more critical approach should be included, on the other hand, the authors should discuss the mentioned theories (from the Introduction) in this section, not just later within the Discussions. Furthermore, the authors should discuss and clarify if their research regards entrepreneurial intentions or intrapreneurial intentions.
The changes your requested have been made in the text. Corrections are highlighted in Red font in the text. Intrapreneurship is emphasized (Line 162-171).
- Regarding section 3 (Materials and Methods), hypothesis H4 and H6 are not sufficiently argued.
The changes your requested have been made in the text. Corrections are highlighted in Red font in the text. (Line 280-291; Line 315-326).
Moreover, within their argument, the Authors should pay attention if they regard intrapreneurial or entrepreneurial intention, considering that respondents are from existing IT companies.
The changes your requested have been made in the text. Corrections are highlighted in Red font in the text. (Line 609-623).
After presenting the research model (Figure 1), the Authors should present the originality elements of their study compared to some other relevant studies.
The changes your requested have been made in the text. Corrections are highlighted in Red font in the text. (Line 346-367).
Concerning the methodological part, the authors should present and argue the used statistical techniques, especially the one to test mediation should be detailed (considering the aim of the study presented witin the abstract), also why is bootstrapping regarded. Eventually some similar studies using these techniques should be referenced.
The changes your requested have been made in the text. Corrections are highlighted in Red font in the text. (Line 418-427).
- Concerning the empirical part, in my opinion, betas from other studies should not be invoked, because samples, research contexts, models perhaps are not entirely comparable.
The changes your requested have been made in the text. Line 494-535.
Moreover, the mediations should be further tested, discussed, not just stating accepted completely or partially.
The changes your requested have been made in the discussion section. Corrections are highlighted in Red font in the text. Line 609-622 and Line 632-650.
- The Conclusions part should be extended, by adding managerial and policy implications, and by extending the research limitations and future research directions.
The changes your requested have been made in the text.
- - if the present study considers no big data, then maybe a motivation to apply Smart-PLS should not refer to this advantage (line 288);
The changes your requested have been made in the text. Line 431-433.
- Authors should review the Acknowledgement part.
The changes your requested have been made in the text. Line 676-677.
- Some sentences should be rephrased: e.g. „Gross Domestic Product (GDP) is expected to be digitalized” (lines 48-49);
The changes your requested have been made in the text. Line 50-52.
“the goodness of fit values of the model is presented” (lines 326-327), etc.
The changes your requested have been made in the text. Line 478.
Thank you for your great support and attention.
Sincerely,
Reviewer 4 Report
Very interesting and pertinent article.
I consider it unprecedented that the authors were able to collect such an expressive sample between a Friday and a Monday, using online and paper questionnaires. In ethical terms, it is recommended to explain how participants are recruited, as well as their inclusion and exclusion criteria. It is not understood whether the 390 participants all belong to one organization/company or several and what the criteria were for selecting the sample. Authors should review the paper and use inclusive language whenever possible. The authors should be clearer by informing readers that they used scales that were not validated for the population and that one of the objectives was also to validate them for the Turkish population. In the Methodology section, it is necessary to better describe the four instruments used, namely, present for each one, the names of the original authors and year, the authors who translated each scale (and year), as well as, describe how many and which the dimensions that make up each scale (questionnaire), the alphas of each dimension (original) and the number of items in each scale, and also present an example of one item per dimension, for each questionnaire used in the study. It is also desirable to indicate which sociodemographic variables were collected (we only have this information when consulting Table 1) and explain the purpose of collecting this information (was it just to characterize the sample?).
The added value of the study must be deepened.
Author Response
Dear Reviewer,
First of all, I would like to express my gratitude for your precious comments and suggestions to improve the article. I took your suggestions into consideration and made the following changes in the article;
I consider it unprecedented that the authors were able to collect such an expressive sample between a Friday and a Monday, using online and paper questionnaires. In ethical terms, it is recommended to explain how participants are recruited, as well as their inclusion and exclusion criteria. It is not understood whether the 390 participants all belong to one organization/company or several and what the criteria were for selecting the sample. Authors should review the paper and use inclusive language whenever possible. The authors should be clearer by informing readers that they used scales that were not validated for the population and that one of the objectives was also to validate them for the Turkish population. In the Methodology section, it is necessary to better describe the four instruments used, namely, present for each one, the names of the original authors and year, the authors who translated each scale (and year), as well as, describe how many and which the dimensions that make up each scale (questionnaire), the alphas of each dimension (original) and the number of items in each scale, and also present an example of one item per dimension, for each questionnaire used in the study. It is also desirable to indicate which sociodemographic variables were collected (we only have this information when consulting Table 1) and explain the purpose of collecting this information (was it just to characterize the sample?).
The added value of the study must be deepened.
The changes your requested have been made in the text. Corrections are highlighted in Red font in the text. Line 369-427. Other changes have been marked in the text in line with the opinions of the other three reviewers.
Thank you for your great support and attention.
Sincerely,
Round 2
Reviewer 3 Report
Dear Authors,
The reviewed manuscript aims to study the mediating effect of innovative behaviour within the influence of digital leadership upon job performance and entrepreneurial intention, testing the research model on a sample of 390 people working in the Turkish IT sector in Istanbul. As stated above, the paper’s theme is interesting, however even after the second round of review, two main aspects should be considered to be improved:
(i) The Authors should discuss in more detail (within the theory and the hypothesis argument) and clarify if their research regards entrepreneurial intentions or intrapreneurial intentions. It is still not clear. Depending on their decision (need to be argued as well), title, hypothesis, methodology, discussion and conclusion.
(ii) Within the empirical part, the mediations (considered within H6 and H7) should be further discussed, not just stating accepted completely or partially. In this sense, what kind of data analysis technique was applied – to be included within the methodological part, motivate why the Authors considered that approach.
I hope the above observations will contribute to the improvement of the reviewed manuscript.
Best regards,
The Reviewer
Author Response
Dear Reviewer,
First of all, I would like to express my gratitude for your precious comments and suggestions to improve the article. I took your suggestions into consideration and made the following changes in the article;
(i) The Authors should discuss in more detail (within the theory and the hypothesis argument) and clarify if their research regards entrepreneurial intentions or intrapreneurial intentions. It is still not clear. Depending on their decision (need to be argued as well), title, hypothesis, methodology, discussion and conclusion.
The changes your requested have been made in the text. The entire article has been re-edited according to Intrapreneurship Intention. Corrections are highlighted in Red font in the text.
(ii) Within the empirical part, the mediations (considered within H6 and H7) should be further discussed, not just stating accepted completely or partially.
The changes your requested have been made in the text. Corrections are highlighted in Red font in the text. Line 250-266, Line 567-576 and Line 589-599
Thank you for your great support and attention.
Sincerely,
Reviewer 4 Report
The authors of the article made a notable effort to include the reviewers' recommendations for improvement. The article now presented is much better and has the quality to be published. Congratulations on the interesting and innovative article.
However, after providing some explanations and including additional information, I consider that some aspects still need to be improved so that the article can proceed to publication:
1. In my opinion, they carried out two studies and not one: a qualitative study - using the survey method through interviews (before obtaining the Opinion of the Ethics Committee - which is not ethical at all); and a quantitative study using the questionnaire survey method - the authors must mention this in the article (they carried out two studies);
2. Regarding study one: it is necessary to include the following information in the article: what is the objective of the interviews; how many subjects were interviewed and what their characteristics were; whether the interviewed subjects were also part of study two or not (and if so, how they proceeded regarding data anonymization and matching); How were the data from study one analyzed (software used)?
Author Response
Dear Reviewer,
First of all, I would like to express my gratitude for your precious comments and suggestions to improve the article. I took your suggestions into consideration and made the following changes in the article;
However, after providing some explanations and including additional information, I consider that some aspects still need to be improved so that the article can proceed to publication:
- In my opinion, they carried out two studies and not one: a qualitative study - using the survey method through interviews (before obtaining the Opinion of the Ethics Committee - which is not ethical at all); and a quantitative study using the questionnaire survey method - the authors must mention this in the article (they carried out two studies);
- Regarding study one: it is necessary to include the following information in the article: what is the objective of the interviews; how many subjects were interviewed and what their characteristics were; whether the interviewed subjects were also part of study two or not (and if so, how they proceeded regarding data anonymization and matching); How were the data from study one analyzed (software used)?
After the reviewer's warning, we realized that we had expressed some things incorrectly. The changes your requested have been made in the discussion section. Corrections are highlighted in Red font in the text.
Line 381-387
Before obtaining ethics permission, 62 businesses with employee numbers ranging from 10 to 280 were visited. By giving information about the questions in the survey form to the businesses which were visited, managers and business owners were informed about the aim of the study and its possible consequences. This information process is definitely a preliminary field research to determine the research areas and it is not possible to have the interviewed people fill out any survey forms and include them in the analysis.
Line 399-404.
“Data belonging to these 390 people were obtained from 21 different businesses. The information provided before obtaining ethical permission was effective in terms of collecting the data for the present study in a short time. Thanks to the information activity, it became easier to meet with managers and businesses that could support the study were determined in this way. This is an obligatory practice in terms of filling out the section related to the research field which is specified in the ethics committee form.”
Thank you for your great support and attention.
Sincerely,